# CFTR Modulation Reduces SARS-CoV-2 Infection in Human Bronchial Epithelial Cells

**DOI:** 10.3390/cells11081347

**Published:** 2022-04-15

**Authors:** Virginia Lotti, Flavia Merigo, Anna Lagni, Andrea Di Clemente, Marco Ligozzi, Paolo Bernardi, Giada Rossini, Ercole Concia, Roberto Plebani, Mario Romano, Andrea Sbarbati, Claudio Sorio, Davide Gibellini

**Affiliations:** 1Microbiology Section, Department of Diagnostic and Public Health, University of Verona, 37134 Verona, Italy; anna.lagni@univr.it (A.L.); andrea.diclemente@univr.it (A.D.C.); marco.ligozzi@univr.it (M.L.); davide.gibellini@univr.it (D.G.); 2Anatomy and Histology Section, Department of Neurosciences, Biomedicine and Movement Sciences, University of Verona, 37134 Verona, Italy; flavia.merigo@univr.it (F.M.); paolo.bernardi@univr.it (P.B.); andrea.sbarbati@univr.it (A.S.); 3Microbiology Unit, IRCCS Azienda Ospedaliero-Universitaria di Bologna, 40138 Bologna, Italy; giada.rossini@aosp.bo.it; 4Department of Diagnostic and Public Health, University of Verona, 37134 Verona, Italy; ercole.concia@univr.it; 5Laboratory of Molecular Medicine, Centre on Advanced Studies and Technology (CAST), Department of Medical, Oral and Biotechnological Sciences, “G. d’Annunzio” University of Chieti-Pescara, 66100 Chieti, Italy; robplebani@gmail.com (R.P.); mromano@unich.it (M.R.); 6General Pathology Section, Department of Medicine, University of Verona, 37134 Verona, Italy; claudio.sorio@univr.it

**Keywords:** SARS-CoV-2 virus, cystic fibrosis, CFTR, ACE-2, CFTR inhibitor, human bronchial epithelial cells

## Abstract

People with cystic fibrosis should be considered at increased risk of developing severe symptoms of COVID-19. Strikingly, a broad array of evidence shows reduced spread of SARS-CoV-2 in these subjects, suggesting a potential role for CFTR in the regulation of SARS-CoV-2 infection/replication. Here, we analyzed SARS-CoV-2 replication in wild-type and CFTR-modified human bronchial epithelial cell lines and primary cells to investigate SARS-CoV-2 infection in people with cystic fibrosis. Both immortalized and primary human bronchial epithelial cells expressing wt or F508del-CFTR along with CRISPR/Cas9 CFTR-ablated clones were infected with SARS-CoV-2 and samples were harvested before and from 24 to 72 h post-infection. CFTR function was also inhibited in wt-CFTR cells with the CFTR-specific inhibitor IOWH-032 and partially restored in F508del-CFTR cells with a combination of CFTR modulators (VX-661+VX-445). Viral load was evaluated by real-time RT-PCR in both supernatant and cell extracts, and ACE-2 expression was analyzed by both western blotting and flow cytometry. SARS-CoV-2 replication was reduced in CFTR-modified bronchial cells compared with wild-type cell lines. No major difference in ACE-2 expression was detected before infection between wild-type and CFTR-modified cells, while a higher expression in wild-type compared to CFTR-modified cells was detectable at 72 h post-infection. Furthermore, inhibition of CFTR channel function elicited significant inhibition of viral replication in cells with wt-CFTR, and correction of CFTR function in F508del-CFTR cells increased the release of SARS-CoV-2 viral particles. Our study provides evidence that CFTR expression/function is involved in the regulation of SARS-CoV-2 replication, thus providing novel insights into the role of CFTR in SARS-CoV-2 infection and the development of therapeutic strategies for COVID-19.

## 1. Introduction

Severe acute respiratory syndrome coronavirus 2 (SARS-CoV-2), a novel transmissible coronavirus, emerged in humans in late 2019 in China [1,2] and caused coronavirus disease 2019 (COVID-19), classified by the WHO as a pandemic [3,4]. The major cellular receptor for SARS-CoV-2 viral entry is cell surface angiotensin-converting enzyme 2 (ACE-2) [5,6], which interacts with the receptor-binding domain (RBD) of the viral S protein [7]. SARS-CoV-2 infection is mainly transmitted by the airway route, and respiratory epithelial cells are the primary targets of the virus, even though ACE-2 is detectable in several human cell types. COVID-19 can either be asymptomatic or elicit a respiratory disease with a wide spectrum of clinical manifestations spanning from mild respiratory symptoms to severe lung injury and multiorgan failure leading to, in the worst cases, death [8]. Interestingly, the clinical severity of SARS-CoV-2 is strongly dependent on the presence of genetic background, concomitant morbidities and the advanced age of patients [9].

Cystic fibrosis (CF) is caused by mutations in the CFTR gene [10] that decrease the number and/or function of CFTR ion channels on the cell membrane, leading to abnormal chloride and bicarbonate secretion. Consequently, an accumulation of a thick and sticky mucus results in inflammation and chronic infections, which are the main culprits of CF-related morbidity and mortality [11,12]. Because of their chronic pulmonary infections, which lead to respiratory failure, people with cystic fibrosis (pwCF) should be considered at high risk of developing severe symptoms of COVID-19 [13]. However, recent studies have indicated that pwCF exhibit a lower clinical impact of SARS-CoV-2 infection, with mild courses of viral disease and no requirement of intensive care unit admission [13,14,15,16].

This intriguing observation regarding the peculiarity of SARS-CoV-2 infection in pwCF drove us to investigate the underlying mechanisms by analyzing the dynamics of SARS-CoV-2 infection in human bronchial epithelial cells, both immortalized and primary cell lines, with a mutated or deleted CFTR gene in comparison with control cell lines.

## 2. Materials and Methods

### 2.1. Cells

Vero E6 cells were cultured in Dulbecco’s modified eagle medium (DMEM, Sigma, Milan, Italy) supplemented with 10% foetal bovine serum (FBS, Euroclone, Milan, Italy), 1% Penicillin/Streptomycin (Gibco, Thermo Fisher, Monza, Italy) [17].

CFBE41o- cells [17,18,19] stably expressing wt-CFTR (CFBE41o- WT) or F508del-CFTR (CFBE41o- ΔF) cDNA were obtained as described previously [20,21]. In brief, CFBE41o- WT were immortalized with SV40 plasmid (pSVori) defective for the origin of replication sequence [18,19] and stably transfected with HIV-based lentiviral vector expressing normal CFTR gene. CFBE41o- ΔF cells were obtained with the same procedure but the HIV-based lentiviral vector expressing the F508del-CFTR gene [20]. CFBE41o- cells were cultured in Minimum Essential Medium (MEM, Gibco, Thermo Fischer) supplemented with 10% FBS (Euroclone), 1% Glutamine (GlutaMAX, Gibco, Thermo Fischer) and a proper amount of selective factor (Puromycin).

16HBE14o- was employed as a control cell line (WT), whereas a clone genetically engineered by CRISPR-Cas9 technology with complete deletion of the CFTR gene (knockout; KO) was used to simulate CFTR nonsense mutations [22,23]. Cells were cultured in MEM (Gibco, Thermo Fischer), supplemented with 10% FBS (Euroclone) and 1% Glutamine (Gibco, Thermo Fischer).

MucilAir™ were purchased from Epithelix Sàrl (Geneva, Switzerland) and maintained in MucilAir™ culture medium (Epithelix Sàrl) at air–liquid interface (ALI) for 1 week upon reception of the products, according to manufacturer’s instruction. MucilAir™ is an in vitro cell model of the human airway epithelium cultured at the air–liquid interface, reconstituted using human primary cells at low passage (P1). To pursue our goal, we have used this fully differentiated bronchial epithelial 3D model from primary human cells derived from healthy donors (*n* = 3; age mean 46 ± 16) and cystic fibrosis patients’ homozygotes ΔF508 (*n* = 3; age mean 28 ± 10).

### 2.2. Viral Strains, Titration and Infection

The SARS-CoV-2 strain was isolated from the respiratory secretions of an adult male patient diagnosed with COVID-19 at S. Orsola Hospital (Bologna, Italy) in March 2020 and replicated in Vero E6 cells as described before [24]. A second SARS-CoV-2 strain (BetaCoV/Australia/VIC01/2020; lot number: 07052020) purchased from NIBSC (London, UK) and isolated from the first patient diagnosed with COVID-19 in Australia [25] was also used.

Viral stocks were titrated by the classic tissue culture infectious dose 50% (TCID50) method in Vero E6 cells as previously described [26,27].

SARS-CoV-2 was inoculated into the cells at a multiplicity of infection (MOI) of 1, and the cells were incubated for 1 h at 37 °C and 5% CO_2_. Supernatant and cell samples were collected for further evaluations before inoculation and every 24 up to 72 h post-infection (hpi). 

For MucilAir™ infection experiments, as reported by Pizzorno et al. [17], the apical sides were gently washed twice with pre-warmed OptiMEM medium (GIBCO, ThermoFisher Scientific) thereafter, 150 μL of SARS-CoV-2 diluted in OptiMEM at MOI = 1 were inoculated and incubated for 1 h at 37 °C and 5% CO_2_. After incubation, the inoculum was removed from the apical side to restore the ALI condition and the media on the basolateral side was changed. Supernatants were collected from apical washes or basolateral medium while cells were harvested in lysis buffer (Promega, Madison, WI, USA) or in RIPA buffer.

#### Cell Treatments

For CFBE41o**^-^**, 10 μM IOWH-032 (MedChemExpress LLC, Monmouth Junction, NJ, USA) were added directly into the flask and incubated for 1h at 37 °C, 5% CO_2_. Thus, after infection (if required), the media was removed, and replaced with fresh media containing 10 μM IOWH-032.

Since their ALI growth, MucilAir™ inserts were treated from the basolateral side. For wt/wt-CFTR MucilAir™, 10 μM IOWH-032 were added and incubated for 1 h at 37 °C, 5% CO_2_ before infection. For F508del/F508del-CFTR MucilAir™, 3 μM VX-661 and 2 μM VX-445 were added and incubated overnight at 37 °C, 5% CO_2_ the day before infection. Both treatments were maintained in chronic for the duration of the experiment.

### 2.3. RT-PCR

The SARS-CoV-2 load was detected with an Allplex 2019-nCoV assay kit (Seegene, Seoul, Korea) following the manufacturer’s instructions. 

RNA was extracted from cells and retrotranscribed into cDNA, which was analyzed by real-time qPCR on a CFX96 Real-Time System (Bio-Rad) using primers indicated in Table 1. Relative gene expression was calculated by the ΔCT method and normalized to the expression of housekeeping control genes. The −ΔΔCt method was used for absolute quantification.

### 2.4. Western Immunoblotting

Total protein lysate was analyzed by SDS-PAGE and transferred to a PVDF membrane (Bio-Rad). Anti-ACE-2 antibody (1:1000, ab15348, Abcam, Cambridge, UK) and/or anti-SARS-CoV-2 (2019-nCoV) spike (S) antibody (1:1000, 40591-MM42, Sino Biological, Wayne, NJ, USA) were probed with the appropriate secondary antibodies (Cell Signaling, Danvers, NJ, USA) and loading control (anti-β-actin antibody; 1:2000, 4970, Cell Signaling). Chemiluminescence signals were acquired by ImageQuantTM LAS 4000 (GE Healthcare Europe GmbH, Milan, Italy).

### 2.5. Flow Cytometry

CFBE41o- WT and ΔF cells were marked with an ACE-2 antibody (1:150 bs-1004R, Bioss, Woburn, MA, USA) and probed with an Alexa Fluor 488-conjugated secondary antibody (10 µg/mL). Analysis was performed using a MACS Quant10 (Miltenyi Biotec GmbH, Bergisch Gladbach, Germany), and data were analyzed using FlowJo™ v.10.7.1. software (BD Life Sciences, Franklin Lakes, NJ, USA).

### 2.6. Immunofluorescence Microscopy

Samples fixed with 4% PFA were incubated overnight at 4 °C with α-ACE-2 antibody (1:200, ab15348, Abcam), reacted with a Cy3 Fab goat α-rabbit antibody (Jackson Laboratory, Baltimore, MD, USA) and stained with TO-PRO 3 (1:3000, Life Technologies, Thermo Fisher). Control samples were prepared by omitting the primary antibody. Images were acquired with a Zeiss LSM 510 confocal microscope (ZEISS Microscopy, Jena, Germany). 

### 2.7. TEER Measurement

Variations in transepithelial electrical resistance (Δ TEER) were measured using a volt-ohm-meter (EVOM2, Epithelial Volt/Ohm Meter for TEER) and STX 2 electrodes (World Precision Instruments, Sarasota, FL, USA), according to the manufacturer’s instruction. Briefly, 200 μL of pre-warmed MucilAir™ culture media were added on the apical side of the insert. After being washed in 70% ethanol, the electrode was equilibrated in saline solution (0.9% NaCl; 1.25 mM CaCl_2_; 10 mM HEPES) until the value of the volt-ohm-meter reached 0.00. For the measurement, the electrode was inserted with the long stem entering through the gap of the insert and leaning on the bottom of the well, and with the short stem dipped in the culture media, above the apical surface. The media were immediately removed from the apical side after the measurement. Values were calculated according to the surface area of the inserts (0.33 cm^2^) and expressed as Ohm·cm^2^. 

### 2.8. Statistical Analysis

All measured parameters were statistically analyzed using GraphPad Prism version 5 (GraphPad software Inc., La Jolla, CA, USA). Comparisons were performed using 2-way ANOVA, and differences were considered statistically significant when *p* < 0.05.

## 3. Results

### 3.1. SARS-CoV-2 Infection Is Less Effective in ΔF Than in WT CFBE41o- Cells

We assessed the impact of SARS-CoV-2 infection on human bronchial epithelial cell lines by analyzing SARS-CoV-2 RNA content in supernatants from 0 to 72 hpi. The SARS-CoV-2 viral load was lower in ΔF than in WT CFBE41o- cells, with a time-dependent trend, as documented by multiplex real-time RT-PCR. In particular, CFBE41o- ΔF cells showed slower SARS-CoV-2 infection than WT cells, with an increase in viral load at 48 hpi, while in CFBE41o- WT cells, the viral RNA copy number was very high at 24 hpi (Figure 1a). To support these findings, we also analyzed the intracellular SARS-CoV-2 content by quantitative real-time RT-PCR targeting the ORF-1ab viral gene. ORF-1ab mRNA expression normalized to GAPDH was significantly different between ΔF and WT CFBE41o- cells (*p* < 0.05) at 48 and 72 hpi (Figure 1b), and the increase in SARS-CoV-2 infection was time-dependent in both cell lines.

To rule out the possibility that this observation was related to a single SARS-CoV-2 strain, we repeated the experiment using an alternative SARS-CoV-2 strain obtained from NIBSC. We confirmed the data described above (Figure 1c), reinforcing the concept that SARS-CoV-2 replication is consistently reduced in CFBE41o- ΔF cells as compared to CFBE41o- WT cells.

Hence, we examined the S protein, which plays a key role in receptor recognition and the cell membrane fusion process, by Western blot analysis of the total lysate targeting the S1 subunit. In accordance with the viral RNA data, the S1 protein was abundantly expressed in CFBE41o- WT samples, especially at 48 and 72 hpi, while it was clearly less abundantly expressed in CFBE41o- ΔF cells over time (Figure 1d).

### 3.2. ACE-2 Expression Is Modulated by SARS-CoV-2 Infection in CFBE41o- Cells

To investigate whether ACE-2 might be related to the reduced SARS-CoV-2 infection in CFTR-defective cells, we performed flow cytometry targeting the ACE-2 protein on the cell membrane in noninfected samples (Figure 2a). The ACE-2 mean fluorescence intensity (MFI) was slightly higher in WT than in ΔF CFBE41o- cells (206 ± 56 and 165 ± 55, respectively; Figure 2b). We also analyzed the ACE-2 protein content in the total protein lysate by Western blotting both before infection and at 24, 48, and 72 hpi. Interestingly, before infection, no major differences were detected in the total ACE-2 protein content in ΔF compared to WT CFBE41o- cells (Figure 2c). Notably, during SARS-CoV-2 infection, ACE-2 expression increased, becoming markedly higher in WT than in ΔF CFBE41o- cells at 72 hpi. These data were supported by mRNA ACE-2 transcription profiling (Figure 2d), which showed a time-dependent increase in ACE-2 mRNA expression following SARS-CoV-2 infection and consistently higher ACE-2 mRNA levels in WT than in ΔF CFBE41o- cells.

Laser-scanning confocal microscopy revealed that ACE-2 immunoreactivity was homogeneously distributed around the nucleus in noninfected CFBE41o- WT cells (Figure 3a). In contrast, in ΔF cells, it was distributed differently and was detectable as spotty staining around the nucleus (Figure 4a). At 24 hpi, these immunoreactivity patterns were enhanced in both cellular subclones infected with SARS-CoV-2 (Figure 3b and Figure 4b, respectively). At 48 hpi, ACE-2 positivity was markedly reduced in the perinuclear area but was evident on the cytoplasmic membrane in WT cells (Figure 3c), while in ΔF cells, it was concentrated in specific areas around the nucleus (Figure 4c). At 72 hpi, ACE-2 immunostaining was predominantly concentrated on the cell membrane in WT cells (Figure 3d) but was granular in appearance and variously distributed throughout the cytoplasm in ΔF cells. Occasionally, it was observed on short stretches of the membrane (Figure 4d). Altogether, these data indicate that the subcellular distribution of ACE-2 is different in WT and CFTR-defective cells.

### 3.3. SARS-CoV-2 Infection Is Strongly Inhibited in Human Bronchial Epithelial Cells with CRISPR-Cas9-Mediated CFTR Gene Deletion

Since ΔF-overexpressing CFBE41o- cells express a mutated, differently processed, but rescuable CFTR function, we performed a SARS-CoV-2 infection challenge using the 16HBE14o- cell line, which was previously shown to permit viral replication without induction of cytopathic effects [29]. Our data confirmed the results obtained with ΔF cells, as the significant inhibition of SARS-CoV-2 replication was even more pronounced when a CFTR-deleted (KO) clone was compared to the isogenic 16HBE14o- WT cell line. A marked delay in the increase in the viral load in the supernatant was supported by a significant difference at 72 hpi (*p* < 0.01) (Figure 5a).

Interestingly, before infection flow cytometry showed higher expression of the ACE-2 membrane protein in WT than in KO 16HBE14o- cells, as indicated by a significant difference (*p* < 0.05) in MFI (150 ± 41 and 94 ± 34, respectively; Figure 5b,c). This result was confirmed by the drastically lower total ACE-2 protein levels in KO than in WT cells as detected by Western blot analysis, demonstrating that complete deletion of the CFTR gene is related to consistent downregulation of ACE-2 protein (Figure 5d).

### 3.4. Pharmacological Inhibition of CFTR Activity Limits SARS-CoV-2 Infection in CFBE41o- WT Cells

As CFTR is involved in the transport of anions, primarily chloride and bicarbonate, we wondered whether its function, rather than its expression, could contribute to the SARS-CoV-2 replication defect highlighted in CF cells. Hence, we used IOWH-032, a synthetic anti-secretory molecule designed to selectively inhibit the CFTR chloride channel selected for being tested in Phase 2 clinical trial as anti-diarrheal [30,31] (Appendix A), on either mock-infected or SARS-CoV-2-infected CFBE41o- WT cells. We compared the supernatant viral loads of WT and ΔF CFBE41o- cells with that of IOWH-032-treated CFBE41o- WT cells at 24, 48 and 72 hpi by multiplex real-time RT-PCR. Remarkably, at 24 hpi, IOWH-032-treated CFBE41o- WT cells showed significantly lower viral loads than untreated WT and ΔF CFBE41o- cells. Moreover, at 72 hpi, the decrease in SARS-CoV-2 viral load in IOWH-032-treated CFBE41o- cells was significant (*p* < 0.0001) even when compared to the load in CFBE41o- ΔF cells (Figure 6a). Preincubation with IOWH-032 did not significantly alter ACE-2 total protein expression in CFBE41o- WT cells before infection, whereas we detected time-dependent increases at 48 and 72 hpi, suggesting that ACE-2 expression is upregulated by SARS-CoV-2 infection (Figure 6b). 

### 3.5. Differences in SARS-CoV-2 Infection and Epithelium Integrity in wt/wt-CFTR and F508del/F508del-CFTR MucilAir™

We replicated the same experimental settings on both wt/wt-CFTR and F508del/F508del-CFTR MucilAir™. SARS-CoV-2 viral production at the epithelial apical surface of wt/wt-CFTR MucilAir™ increased sharply at 48 hpi, with the highest virus titers (>1 × 10^9^ copies/µL) being observed by 72 hpi while, in comparison, the quantity produced by F508del/F508del-CFTR MucilAir™ results significantly reduced at 48 and 72 hpi (*p* < 0.05 and *p* < 0.0001, respectively), with no increment during the time (Figure 7a). The sharp increase in wt/wt-CFTR MucilAir™ viral replication correlated with a reduction in epithelium integrity at 48 but mostly at 72 hpi, reflected by 1.2- and 2.9-fold decreases in TEER values, respectively, while in F508del/F508del-CFTR MucilAir™ the TEER values remain consistent with the passing of hours after infection (1.4- fold decrease at 72 hpi) stating greater maintenance of epithelium integrity (Figure 7b). Viral genome was detected in the basal medium at 48 hpi and onward (Figure 7c), in agreement with SARS-CoV-2 detection at the apical side, strongly present in wt/wt-CFTR while weakly detected in F508del/F508del-CFTR even at 72 hpi, indirectly confirming the breach of epithelial integrity caused by the infection more pronounced in wt/wt-CFTR than in F508del/F508del-CFTR MucilAir™, previously highlighted through TEER measurements.

### 3.6. Drug Modulation of CFTR Able to Improve or Worsen SARS-CoV-2 Infection in In Vitro 3D Human Airway Epithelium

To investigate whether CFTR could be involved in this mechanism of defective replication we tested both the VX-661+VX-445 treatment, a well-known highly effective next-generation CFTR modulator drug, acting by helping to process misfolded CFTR protein to the cell membrane, and increasing channel opening [32,33], in F508del/F508del-CFTR MucilAir™, and IOWH-032, a CFTR-selective inhibitor, in wt/wt-CFTR MucilAir™. As shown in Figure 8a, wt/wt-CFTR MucilAir™ treated with CFTR-inhibitor IOWH-032 revealed a lower SARS-CoV-2 replication at 24 hpi compared to wt/wt-CFTR not treated MucilAir™, which is maintained at 48 and consolidated at 72 hpi (*n* = 3; *p* < 0.01). Interestingly the recovery of CFTR expression and function by VX-661+VX-445 treatment in F508del/F508del-CFTR MucilAir™ is associated with a sudden increase in viral particles from 48 to 72 hpi if compared to non-treated F508del/F508del-CFTR that is particularly evident at 48 hpi. Moreover, TEER measurements revealed that the effects induced by IOWH-032 on wt/wt-CFTR MucilAir™ also translated into protection of the barrier integrity at 72 hpi, while VX-661+VX-445 treatment evidenced on F508del/F508del-CFTR MucilAir™ a partial recovery of cell-to-cell integrity at 48 hpi (*p* < 0.0001) that is essentially lost at 72 hpi, in line with the capability of the treatment to partially recover CFTR function and in accordance with the increase in SARS-CoV-2 viral particles’ production (Figure 8b).

## 4. Discussion

In this study, we analyzed SARS-CoV-2 infection, comparing wt/wt with CFTR-modified human bronchial epithelial cells. Our results indicated that (i) SARS-CoV-2 infection is significantly inhibited in human bronchial epithelial cells in which the CFTR gene is completely deleted or dysfunctional; (ii) human bronchial epithelial cell lines with targeted deletion of the CFTR gene show stronger basal downregulation of ACE-2 expression in comparison to the CFBE41o- ΔF cell line, in which only a slight downregulation is observed; and (iii) the decrease in viral replication with CFTR gene mutation or deletion is related to the loss of CFTR function, as confirmed by treatment with IOWH-032, an inhibitor of CFTR, impairing the viral replication cycle, and by VX-661+VX-445 treatment, which in turn improves SARS-CoV-2 viral replication in CFTR-modified cells, partially restoring CFTR functionality.

Different studies have reported that the severity, number of infection cases and viral spread of SARS-CoV-2 in pwCF are significantly lower than those in the normal population [34,35], even though pwCF are expected to be at increased risk of developing severe manifestations of COVID-19 since they are clear examples of “fragile” patients. In the present study, the in vitro analysis of viral load demonstrated that alteration or complete deletion of the CFTR gene elicited a significant decrease in SARS-CoV-2 content in both the cellular supernatant and within the cells. Interestingly, the highest inhibition of viral replication was observed when CFTR was completely deleted, suggesting that the impact of SARS-CoV-2 infection might be dependent on a complete loss of CFTR expression/activity. Consistent with the observation of decreased replication, we also found that CFTR-mutated cells showed lower expression of the ORF1-ab gene than WT cells, suggesting decreased expression of non-structural proteins that are essential for the creation of viral replication–transcription complexes (RTCs) or replication organelles (ROs) [36]. In accordance with these data, the results obtained on MucilAir™ showed a lower number of viral particles in F508del/F508del-CFTR compared to wt/wt-CFTR over time, correlated with a minor loss of integrity of epithelium recorded by TEER measurements.

Recent studies have suggested that several factors are protective against SARS-CoV-2 infection in pwCF [37,38] and that mutations in the CFTR gene, due to pH changing of organelles in the protein secretory pathway, may alter the protein abundance and/or the glycosylation pattern of ACE-2 and/or TMPRSS-2 and consequently mitigate the effects of SARS-CoV-2 infection [13,39]. Moreover, previous studies have reported that cellular ACE-2 expression in the nasal epithelium is lower in pediatric patients than in adult patients, suggesting a possible mechanism to explain their asymptomatic or mild symptomatic clinical evolution of SARS-CoV-2 infection [40]. In our study, the level of cell membrane ACE-2 expression was slightly lower in ΔF than in WT CFBE41o- cells, indicating a possible role of ACE-2 in the reduction in SARS-CoV-2 viral load linked to host cell entry. This was associated with a diverse subcellular ACE-2 distribution pattern observed by immunofluorescence analysis between the two CFBE41o- cell lines under basal conditions; perinuclear localization of ACE-2 was slightly more pronounced in WT than in ΔF CFBE41o- cells, even at 24 hpi. At 48 and 72 hpi, ACE-2 immunostaining was predominantly localized on the cell membrane in WT cells, while in CFBE41o- ΔF cells, it was primarily localized in the cytoplasm and only occasionally observed on the cell membrane. These data confirm that ACE-2 expression differs in the two subtypes of CFBE41o- cells dependent on the time of infection, which could suggest alterations in the steps of coronavirus internalization and/or replication. In addition, immunoblot analysis evidenced slightly lower total ACE-2 protein expression in the ΔF than in the WT CFBE41o- cell line at 72 hpi. Interestingly, under basal conditions, no major differences were detectable in total ACE-2 protein expression between the WT and ΔF CFBE41o- cell lines. Notably, we found higher expression of total ACE-2 protein in WT than in KO 16HBE14o- cells under basal conditions, suggesting a possible link between the lack or dysfunction of CFTR ion channels and the regulation of ACE-2 membrane expression. The variable expression pattern of ACE-2 in respiratory epithelia and the complex role of ACE-2 could indicate that ACE-2 expression levels and patterns might not be the only factors affecting the susceptibility of cells to the virus. The decrease in ACE-2 may partially explain the decrease in SARS-CoV-2 infection, but it may also suggest that intracellular conditions play important roles; for example, CFTR mutation-induced pH alterations in organelles of the secretory pathway may alter glycosylation of the ACE-2 receptor [41], mitigating viral infection and the damaging effects of COVID-19 on the lungs [13]. In CFTR-inhibited CFBE41o- WT cells ACE-2 expression seems to be upregulated over time by SARS-CoV-2 infection. Nonetheless, in the same experimental condition, the viral load is decreased even compared to that in CFBE41o- ΔF cells, suggesting that functional alteration of CFTR and the consequent ionic imbalance are key factors affecting viral replication, above expression of classical SARS-CoV-2 ACE-2 receptor.

These findings could also be strengthened by results obtained by CFTR modulation. Thus, wt/wt-CFTR primary cells with inhibited CFTR showed a drop in SARS-CoV-2 viral load, suggesting a pivotal role of CFTR function in SARS-CoV-2 infection. Furthermore, F508del/F508del-CFTR MucilAir™, treated with VX-661+VX-445, after a partial recovery of CFTR function and expression of a properly glycosylated form, showed an increment of SARS-CoV-2 viral load when compared to untreated cells. Notably, the alteration of ionic regulation may lead to disruption of the viral replication cycle and intracellular changes in pH that produce significant alterations in protein assembly and structure. A recent study reported that the SARS-CoV-2 S protein is activated under acidic conditions by the endolysosome proteases TMPRSS-2 and cathepsins B and L, so deacidification of this organelle has been shown to deactivate proteases and suppress viral infection [5]. Moreover, several studies have reported a role for CFTR in the trafficking and fusion of endosomes [42], the main route of entry into the cell for many enveloped viruses, including SARS-CoV-2. Panou and colleagues [43] reported that the blockade of CFTR in BK polyomavirus (BKPyV)-infected primary kidney cells significantly reduces the transport of virions to the ER [44,45], supporting the hypothesis that impairment of CFTR function and related imbalances could play pivotal roles in viral replication.

Taken together, our data indicate that SARS-CoV-2 infection is lower in CFTR-defective cells than in normal cells, although the mechanism(s) underlying this observation should be better elucidated. Viral replication inhibition in CFTR-defective cells may be related to several mechanisms, including ACE-2 regulation. However, data obtained after CFTR ablation and specific CFTR inhibitor treatment indicate that ionic dysregulation due to loss of CFTR function might be a key mechanism, providing new insights into SARS-CoV-2 infection and possible new pharmacological avenues.

## Figures and Tables

**Figure 1 cells-11-01347-f001:**
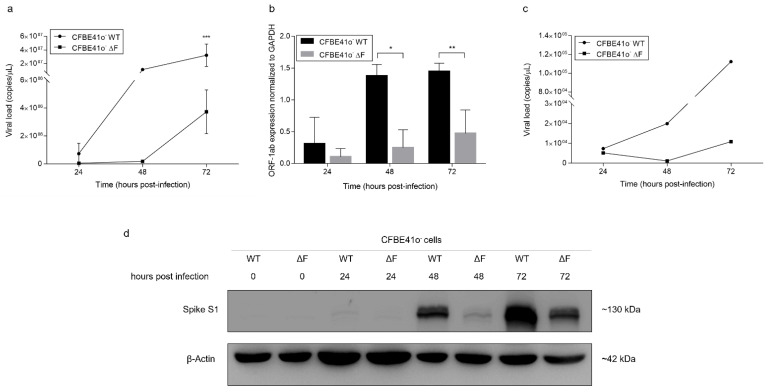
SARS-CoV-2 replication capability is different in WT and ΔF CFBE41o- cells: (**a**) Time course measurement of viral load in the supernatant of CFBE41o- cells infected with SARS-CoV-2 by commercial qualitative real-time RT-PCR (Seegene) able to detect several viral gene targets. The data are presented as the mean ± SD from independent experiments (*n* = 4; *** *p* < 0.001). (**b**) Measurement of ORF-1ab mRNA expression normalized to GAPDH (used as a housekeeping control) in cells infected with RNA extracted from SARS-CoV-2-infected CFBE41o- cells. The data are presented as the mean ± SD from independent experiments (*n* = 3; * *p* < 0.05, ** *p* < 0.01). (**c**) Time course measurement of viral load in the supernatant of WT and ΔF CFBE41o- cells infected with the BetaCoV/Australia/VIC01/2020 SARS-CoV-2 strain. The data were obtained from semiquantitative PCR (Seegene) of cellular supernatant and are expressed as the copies/µL ratio. (**d**) Total S1 protein quantification by Western blot analysis in CFBE41o- cells. Samples analyzed were harvested from 0 to 72 hpi with SARS-CoV-2. β-Actin was used as a loading control. The blot is representative of *n* = 3.

**Figure 2 cells-11-01347-f002:**
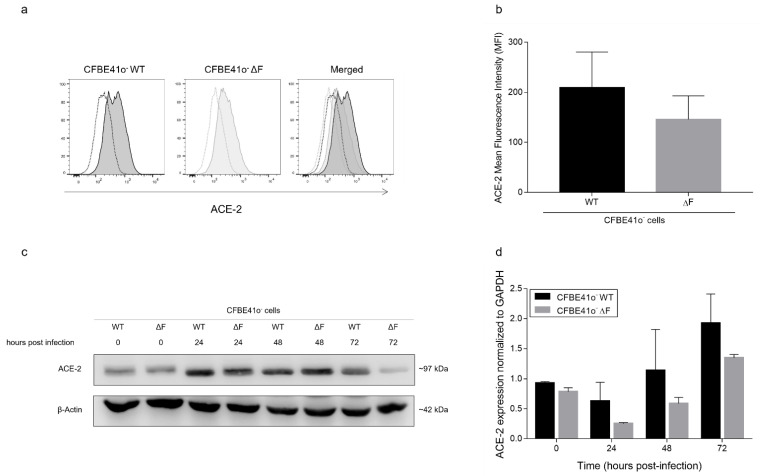
ACE-2 expression levels in CFBE41o- cells determined by different techniques: (**a**) Flow cytometry of ACE-2 cell membrane expression in uninfected ΔF and WTCFBE41o- cells. The data are representative of *n* = 3. (**b**) MFI of ACE-2 in uninfected ΔF and WT CFBE41o- cells. The data are presented as the mean ± SD from independent experiments (*n* = 3). (**c**) Total ACE-2 protein quantification by Western blot analysis in CFBE41o- cells. The samples analyzed were harvested from 0 to 72 hpi with SARS-CoV-2. β-Actin was used as a loading control. The data are from a Western blot representative of *n* = 3. (**d**) Measurement of ACE-2 mRNA expression normalized to GAPDH (used as a housekeeping control) in RNA extracted from SARS-CoV-2-infected CFBE41o- cells. The data are presented as the mean ± SD from independent experiments (*n* = 3).

**Figure 3 cells-11-01347-f003:**
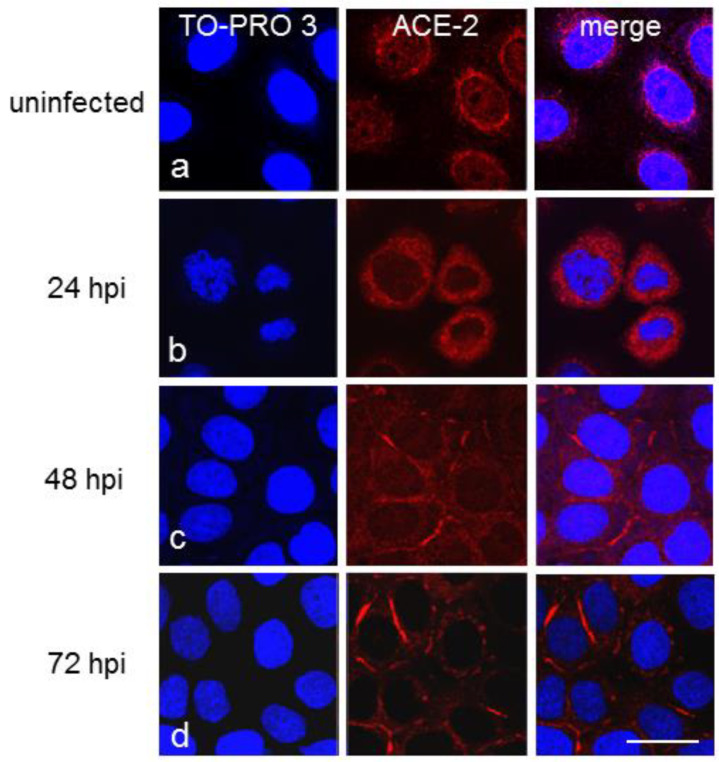
Immunofluorescence confocal microscopy showing the expression of ACE-2 (red) in uninfected (**a**) and infected CFBE41o- WT cells obtained at 24 (**b**), 48 (**c**), 72 (**d**) hpi. Cell nuclei were counterstained with TO-PRO 3 (blue). Scale bar: 20 µm.

**Figure 4 cells-11-01347-f004:**
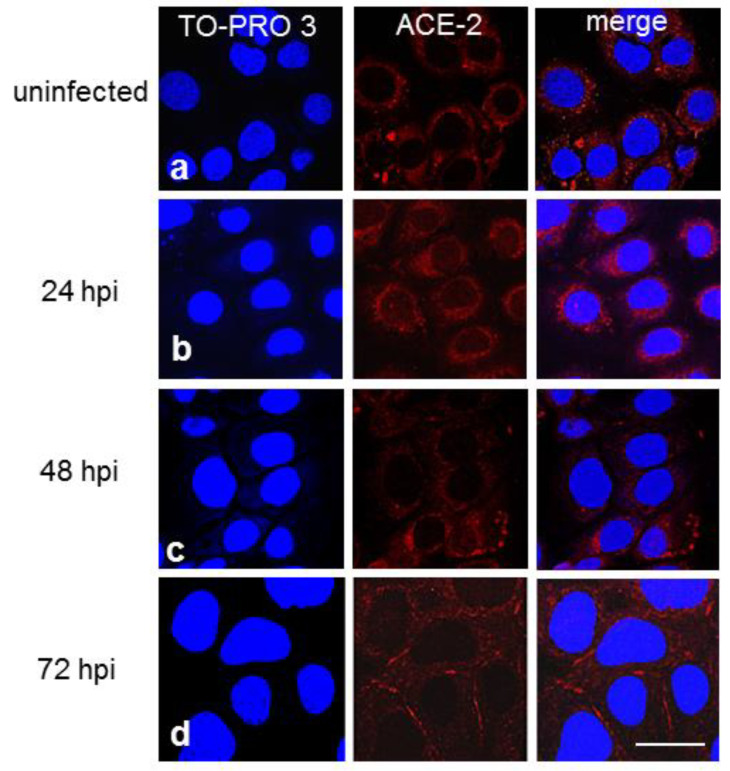
Immunofluorescence confocal microscopy showing the expression of ACE-2 (red) in uninfected (**a**) and infected CFBE41o- ΔF cells obtained at 24 (**b**), 48 (**c**), 72 (**d**) hpi. Cell nuclei were counterstained with TO-PRO 3 (blue). Scale bar: 20 µm.

**Figure 5 cells-11-01347-f005:**
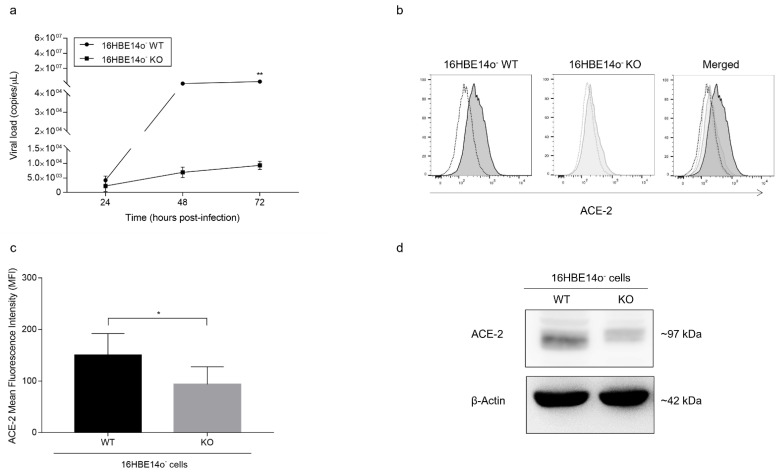
SARS-CoV-2 replication and ACE-2 expression in WT and KO 16HBE14o- cells: (**a**) Time course measurement of viral load in the supernatant of 16HBE14o- cells infected with SARS-CoV-2 analyzed by qualitative real-time RT-PCR (Seegene) able to detect several viral gene targets. The data are presented as the mean ± SD from independent experiments (*n* = 4; ** *p* < 0.01). (**b**) Flow cytometry analysis of ACE-2 cell membrane expression in uninfected WT and KO 16HBE14o- cells. The data are representative of *n* = 3 measurements. (**c**) MFI of ACE-2 in uninfected WT and KO 16HBE14o- cells. The bars represent the mean ± SD from independent experiments (*n* = 3; * *p* < 0.05). (**d**) Total ACE-2 protein quantification by Western blot analysis in 16HBE14o- cells. The samples analyzed were harvested from 0 to 72 hpi with SARS-CoV-2. β-Actin was used as a loading control. The data are representative of *n* = 3.

**Figure 6 cells-11-01347-f006:**
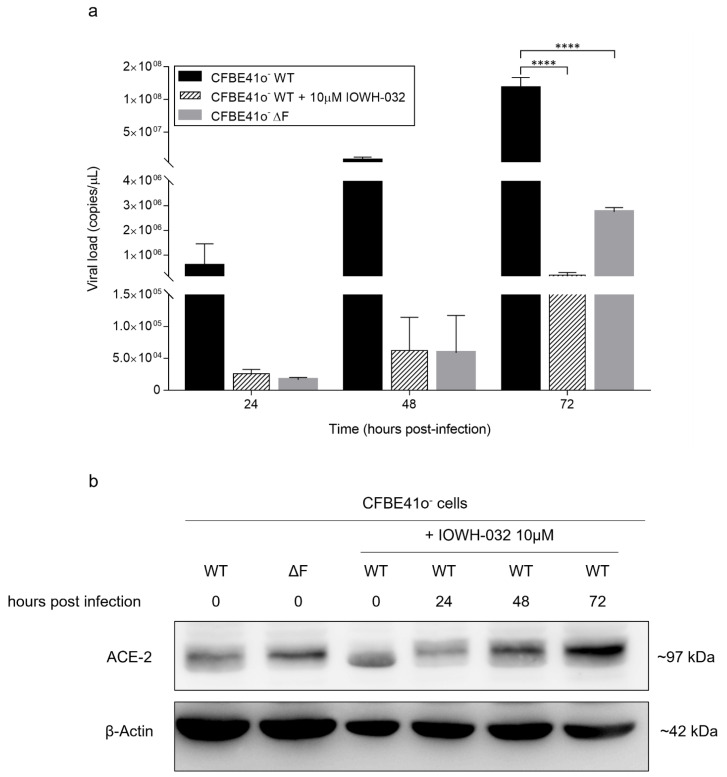
Effect of CFTR inhibition in CFBE41o- WT cells via IOWH-032 treatment compared to WT and ΔF CFBE41o- cells: (**a**) Time course measurement of viral load in the supernatant of CFBE41o- cells infected with SARS-CoV-2 analyzed by qualitative real-time RT-PCR (Seegene). The data are presented as the mean ± SD from independent experiments (*n* = 3; **** *p* < 0.0001). (**b**) Total ACE-2 protein quantification by Western blot analysis in CFBE41o- cells. Samples were harvested from CFBE41o- WT cells exposed or not to IOWH-032 as well as CFBE41o- ΔF cells at different time points of SARS-CoV-2 infection. β-Actin was used as a loading control. The data are representative of *n* = 3 experiments.

**Figure 7 cells-11-01347-f007:**
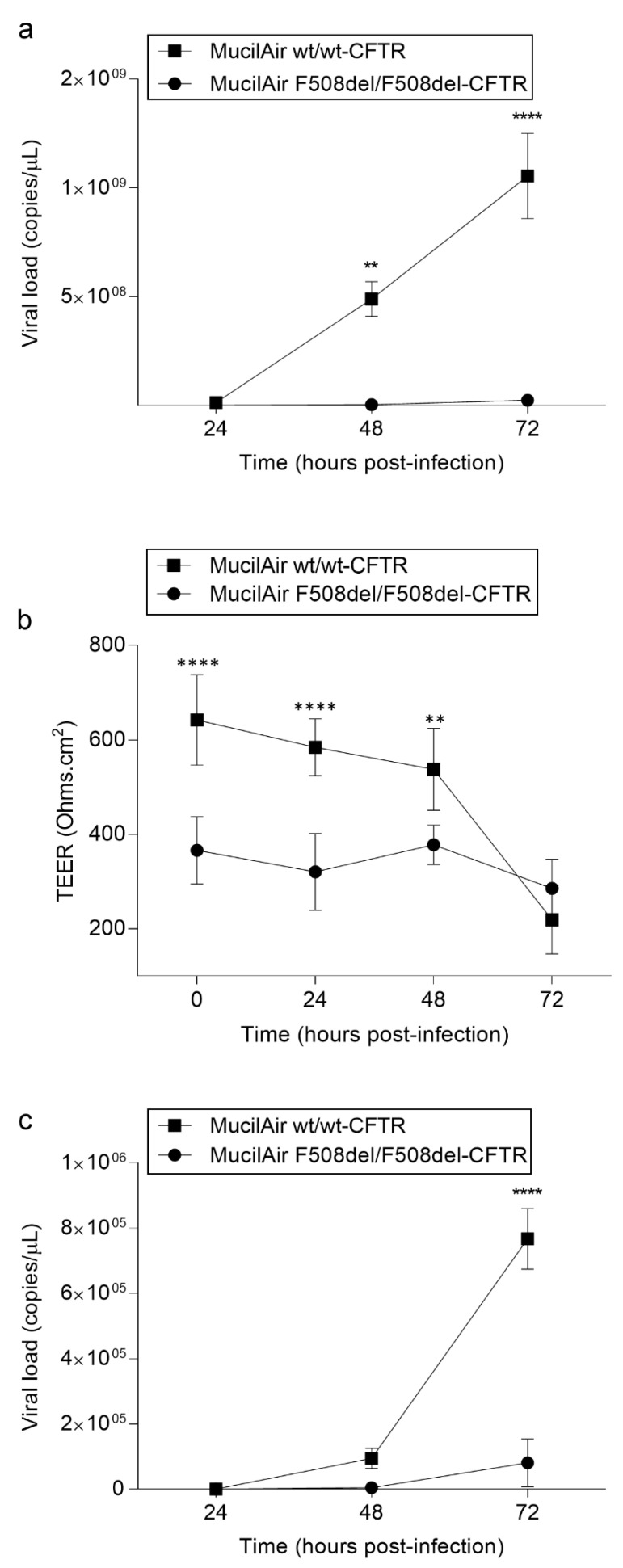
SARS-CoV-2 replication in wt/wt-CFTR and F508del/F508del-CFTR MucilAir™: (**a**) Apical viral production was assessed in washes of the apical side at 24, 48, 72 hpi of both wt/wt-CFTR and F508del/F508del-CFTR MucilAir™ by commercial qualitative real-time RT-PCR (Seegene). The data are presented as the mean ± SD from independent experiments (*n* = 3; ** *p* < 0.01; **** *p* < 0.0001). (**b**) Trans-epithelial resistance measurements in wt/wt-CFTR and F508del/F508del-CFTR MucilAir™ (TEER in Ω·cm^2^) between the apical and basal sides at each time point. The data are presented as the mean ± SD from independent experiments (*n* = 3). (**c**) Basolateral SARS-CoV-2 quantification in wt/wt-CFTR and F508del/F508del-CFTR MucilAir™. Basolateral SARS-CoV-2 viral quantification was assessed in the basolateral medium at 24, 48, 72 hpi of both wt/wt-CFTR and F508del/F508del-CFTR MucilAir™ by commercial qualitative real-time RT-PCR (Seegene) able to detect several viral gene targets. The data are presented as the mean ± SD from independent experiments (*n* = 3; **** *p* < 0.0001).

**Figure 8 cells-11-01347-f008:**
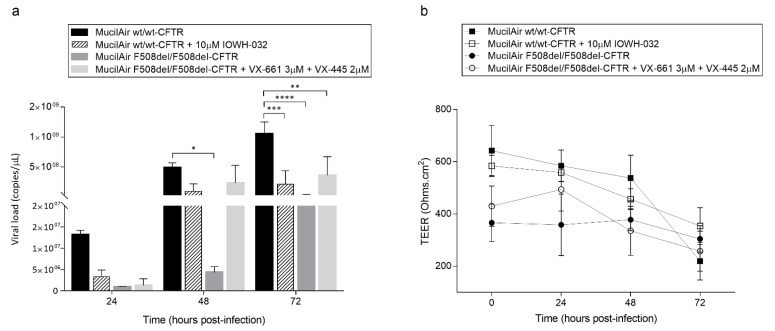
SARS-CoV-2 replication cycle in MucilAir™ is driven by CFTR modulation: (**a**) Apical viral production was assessed in washes of the apical side at 24, 48, 72 hpi of both wt/wt-CFTR and F508del/F508del-CFTR MucilAir™ with and without CFTR-modulator/inhibitor treatment by commercial qualitative real-time RT-PCR (Seegene). The data are presented as the mean ± SD from independent experiments (*n* = 3; * *p* < 0.05; ** *p* < 0.01; *** *p* < 0.001; **** *p* < 0.0001). (**b**) Trans-epithelial resistance (TEER in Ω·cm^2^) between the apical and basal sides was measured at each time point in both wt/wt-CFTR and F508del/F508del-CFTR MucilAir™ with and without VX-661+VX-445, for F508del/F508del-CFTR, or IOWH-032, for wt/wt-CFTR, chronic treatments. The data are presented as the mean ± SD from independent experiments (*n* = 3).

**Table 1 cells-11-01347-t001:** Sequences of primers used for qRT-PCR [28].

Primers	Sequences (5′ -> 3′)
ORF1ab F	TGATGATACTCTCTGACGATGCTGT
ORF1ab R	CTCAGTCCAACATTTTGCTTCAGA
ACE-2 F	AAACATACTGTGACCCCGCAT
ACE-2 R	CCAAGCCTCAGCATATTGAACA
GAPDH F	TCAAGAAGGTGGTGAAGCAGG
GAPDH R	CAGCGTCAAAGGTGGAGGAGT
β-Actin F	CCCTGGACTTCGAGCAAGAG
β-Actin R	ACTCCATGCCCAGGAAGGAA

Abbreviations: F, forward; GAPDH, glyceraldehyde 3-phosphate dehydrogenase; R, reverse.

## Data Availability

Not applicable.

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
