# Peer review of "CFTR Modulation Reduces SARS-CoV-2 Infection in Human Bronchial Epithelial Cells"

_cells, 2022, doi:10.3390/cells11081347_

Round 1

Reviewer 1 Report

I am recommending rejection of this publication due to significant  academic fraud and methodological flaws

The academic fraud is the re-use of western blots to represent different results. The full western blot for figure 2 is the same as the western blot for figure 5/6, just re-imaged at a different exposure. This can be seen by overlaying the images in a program such as Photoshop, ImageJ or GIMP. Lanes 9/10 from fig 2 have been removed in fig 5/6.

Author Response

The accusation of academic fraud turns out to be really heavy and in this specific case also extremely misplaced. The thesis that the two figures show the same blot with different exposures is refuted by comparing the original images sent in tiff format. However, we realized that we had erroneously added the same anti-actin blot to the original figures 1 and 5-6, a mistake we recognize and apologize for.

Please, see the attached file for the complete rebuttal.

Reviewer 2 Report

The manuscript by Lotti et al explores the impact of CFTR function in susceptibility on SARS-CoV2 replication.  This manuscript is very well written and clearly shows by multiple routes that CFTR function is involved in regulating the replication efficiency in epithelial cells. 

Minor comments

  1. In figure 6, it would be more informative if side-by-side comparisons were made with WT not treated with CFTR inhibitor and F508del over time to better evaluate the impact of the IOWH-032.
  2. In figure 8, a more direct analysis of CFTR correction in F508del cells by vx-661 and vx-445 by Ussing chamber analysis would be beneficial in assessing infection data. Is restored function 10%, 50% of WT?  How much CFTR function is needed?

Overall, this is an excellent manuscript that sheds light on the role of CFTR in viral protection.  Further studies on the mechanisms need to be accomplished, but not for this manuscript specifically.  There was data by the Erzurum group that lack of IFN-gamma/STAT1 signaling impacted viral susceptibility that could be discussed in this context. 

Author Response

Minor comments:

  1. In figure 6, it would be more informative if side-by-side comparisons were made with WT not treated with CFTR inhibitor and F508del over time to better evaluate the impact of the IOWH-032.

We rearranged the graph in figure 6, as suggested. Unfortunately, we do not have any WB membrane with the lane order you indicate and then it is useful to leave unchanged the original blots. However, we are ready to do so if deemed appropriate by the reviewer.

  1. In figure 8, a more direct analysis of CFTR correction in F508del cells by vx-661 and vx-445 by Ussing chamber analysis would be beneficial in assessing infection data. Is restored function 10%, 50% of WT?  How much CFTR function is needed?

The recovery is close to 50%. We indicated this data in Supplementary Figure 2.

Please, see the attached file for the complete rebuttal.

Reviewer 3 Report

I have some specific comments for the authors:

- Abstract needs to be re-written as it is not clear and structured properly. Start with a brief intro, the aim of the study, methods used and important findings of the study.

- CFTR-modified bronchial epithelial cells with what? It is not clear from the sentence. Please emphasise in the Abstract that these findings were validated in the primary epithelial cells.

- Line 48-51, make short sentences.

- In the methods section, briefly mention the treatment with CFTRinh172 and correctors as its missing; also, note the preincubation with the correctors and inhibitor whether it is done on the apical or basal side?

- Change the “pole” word throughout the manuscript to “side” or “compartment.”

- TEER measurement: change “Ω/cm2 to “Ω.cm2” change throughout the manuscript.

- Line 226-227: massively less - was it significant?

- It would be ideal if the authors also show western blot for CFTR, i.e., band B and band C and if there is any change in the protein expression levels post-COVID-19 infection in both wt/wt and F508del/F508del cells.

- Revisit lines 325 and 326 of the discussion are not very clear.

- Why IOW-32 inhibitor is not widely used in the CF field. Is there any specific reason for the use of IOW-32? Did the authors try other CFTR inhibitors (CFTRinh172 and GlyH101) apart from IOW-32?

- Line 390-394, many sentences were clubbed together, and it’s unclear what message the authors want to convey to the reader.

- Authors have only shown the ion transport data with inhibitor usage (toxicity). It would be ideal if they also represent after correctors + virus preincubation/SARS-CoV infection, just to strengthen what they claim about CFTR involvement.

- F508del+/+ can be changed to F508del/F508del-CFTR and wt/wt-CFTR for primaries and for stably expressing cell lines.

Author Response

I have some specific comments for the authors:

- Abstract needs to be re-written as it is not clear and structured properly. Start with a brief intro, the aim of the study, methods used and important findings of the study.

We have re-written the abstract, as suggested.

- CFTR-modified bronchial epithelial cells with what? It is not clear from the sentence. Please emphasise in the Abstract that these findings were validated in the primary epithelial cells.

We added in the material and method section the description of the modification of CFBE41o- cells. In addition, we emphasized that our results were validated in primary cells.

- Line 48-51, make short sentences.

We shortened the sentences.

- In the methods section, briefly mention the treatment with CFTRinh172 and correctors as its missing; also, note the preincubation with the correctors and inhibitor whether it is done on the apical or basal side?

We added a new section in Material and methods (“Cell treatments”) where we specified that the pretreatments of cells with CFTR-correctors and inhibitor, was carried out in the basolateral side in MucilAir™ inserts.

- Change the “pole” word throughout the manuscript to “side” or “compartment.”

We performed the suggested change.

- TEER measurement: change “Ω/cm2 to “Ω.cm2” change throughout the manuscript.

We corrected the measurement unit.

- Line 226-227: massively less - was it significant?

Yes, the difference was statistically significant, and we changed the sentence in the text to be clearer.

- It would be ideal if the authors also show western blot for CFTR, i.e., band B and band C and if there is any change in the protein expression levels post-COVID-19 infection in both wt/wt and F508del/F508del cells.

We added the WB suggested as Supplementary Figure 3.

- Revisit lines 325 and 326 of the discussion are not very clear.

We amended the specific sentence to improve the clarity of the message.

- Why IOW-32 inhibitor is not widely used in the CF field. Is there any specific reason for the use of IOW-32? Did the authors try other CFTR inhibitors (CFTRinh172 and GlyH101) apart from IOW-32?

We preferred the use of IOWH-032 since this molecule results in phase 2 clinical trial (https://clinicaltrials.gov/ct2/show/NCT02111304) and might be considered for clinical application to reduce SARS-CoV-2 replication capability. We tried other CFTR-inhibitors (such as inh172) but in a single experiment since we were mainly focused on IOWH-032. 

- Line 390-394, many sentences were clubbed together, and it’s unclear what message the authors want to convey to the reader.

We amended the specific sentence and spilt in shorter sentences, as requested.

- Authors have only shown the ion transport data with inhibitor usage (toxicity). It would be ideal if they also represent after correctors + virus preincubation/SARS-CoV infection, just to strengthen what they claim about CFTR involvement.

Unfortunately, at our knowledge, it is not possible for safety reasons. The local regulations prohibited the manipulation of SARS-CoV-2 or SARS-CoV-2 infected cells outside BSL-3 laboratory. Thus, we will not be able to perform Ussing chamber assay on infected cells and we cannot fix the cells for safety since we do need live cells to perform electrophysiological assays.

- F508del+/+ can be changed to F508del/F508del-CFTR and wt/wt-CFTR for primaries and for stably expressing cell lines.

We carried out the suggested change for the primary cells. However, for the CFBE41o- cells we propose to follow the nomenclature used by Bebok et al., being these cells transfected with cDNA. Corrections have been made throughout the text, figures, and supplementary materials.

(Bebok, Z.; Collawn, J.F.; Wakefield, J.; Parker, W.; Li, Y.; Varga, K.; Sorscher, E.J.; Clancy, J.P. Failure of cAMP agonists to activate rescued deltaF508 CFTR in CFBE41o- airway epithelial monolayers. J Physiol 2005, 569, 601-615, doi:10.1113/jphysiol.2005.096669.)

Please, see the attached file for the complete rebuttal.

Round 2

Reviewer 3 Report

I have some minor comments:

It would be interesting to use Bumetanide and block the NKCC cotransporter on the basal side and then do the studies to check the involvement of Cl- secretion and, in turn, CFTR.

It would be ideal if the Authors also add/include other inhibitors data in the supplementary data and mention briefly in the discussion that we have also tested other CFTR and we found that --

Are there any other studies that have used IOWH-032 in viral studies? On what basis are the authors claiming that it might be considered to reduce SARS-CoV-2?

In cell treatment, it’s still not mentioned how much volume was added to the apical side?

Author Response

It would be interesting to use Bumetanide and block the NKCC cotransporter on the basal side and then do the studies to check the involvement of Cl- secretion and, in turn, CFTR.

Thank you for the suggestion but please remind that the results of the Ussing chamber experiments were presented with the sole purpose of demonstrating the efficacy of IOWH-032 in selectively closing the CFTR channel. We have used standard procedures including Amiloride at the beginning of the experiment to avoid EnaC interference and to measure chloride secretion mediated by CFTR. It could be interesting to also use Bumetanide (along with other Cl- secretion channels) but we think that in this case it does not fits with the scope of study that primarily address the question whether CFTR alterations are in some way involved in viral replication and whether a CFTR inhibition strategy might be of use in this scenario.

It would be ideal if the Authors also add/include other inhibitors data in the supplementary data and mention briefly in the discussion that we have also tested other CFTR and we found that –

Indeed, this is an interesting point to develop. We are preparing another manuscript extending the type of drugs and addressing the role of CFTR-inhibitors in SARS-CoV-2 infection. At the moment, we do not have enough data to be presented and discussed.

Are there any other studies that have used IOWH-032 in viral studies? On what basis are the authors claiming that it might be considered to reduce SARS-CoV-2?

The drug was used for application as anti-diarrhea and we have not retrieved other studies targeting CFTR as anti-viral, including of course IOWH-032 itself. As discussed along the paper, we suggest that IOWH-032, and by extension any other selective CFTR inhibitor, might play a role in modulating SARS-CoV-2 replication in WT cells by blocking CFTR function. Of course, this is the first study of this kind and further studies need to be planned to deepen our knowledge in this field.

In cell treatment, it’s still not mentioned how much volume was added to the apical side?

The volume added in the apical side for the inoculum with the virus is 150 μL, information You can find in material and methods section, paragraph 2.2 “Viral strains, titration and infection” (line 112 in the revised version “cells-1630888”).
